# Development of a context-specific trauma scale among a Palestinian adult population living amidst military violence

Fayez Mahamid[1] 🆔, Guido Veronese[2] 🆔 and Dana Bdier[1,2]

[1]Psychology and Counseling Department, An-Najah National University, Nablus, Palestine and [2]Department of Human Sciences and Education, University of Milano-Bicocca, Milan, Italy

context-specific trauma; political trauma; test development; military violence; Palestine

**Corresponding author:**
Fayez Mahamid;
Email: mahamid@najah.edu

## Abstract

The current study aimed to develop a context-specific trauma scale in the Palestinian context. The sample of our study consisted of 490 Palestinian adults – 230 males and 260 females. Our scale ended up with 32 items to measure traumatic symptoms in the Palestinian context. Results of exploratory factor analysis and confirmatory factor analysis revealed a stable construct of a five-factor structure of the Palestinian specific-context trauma: (1) re-experiencing trauma, (2) avoidance and numbing, (3) hyperarousal, (4) somatic symptoms and (5) psychological symptoms. Reliability of the scale was further established by assessing the test–retest and internal consistency of all subscales. Convergent validity for the context-specific trauma scale was conducted by testing the association between the scale and two existing measures – the WHOQOL-BREF and the Impact of the Event Scale (IES-R). We recommend using our scale in empirical studies incorporating spoken or written disclosure about traumatic experiences. The scale should also be considered when working with clinical and non-clinical groups who have experienced politics-related trauma.

## Impact statement

Palestinian people have been facing humanitarian disaster and trauma since 1948. This community urgently requires intensive mental health interventions to help individuals effectively cope with ongoing traumatic events. Developing and validating new measures to assess traumatic symptoms in the Palestinian context will help mental health professionals provide therapeutic and supportive services to those who are continually at risk of developing trauma and other psychological disorders.

## Introduction

Since the 1948 war, Palestine has been under military occupation by Israel (Marie et al., 2020). Since the event known as *al-Nakba* (the catastrophe) that occurred 74 years ago, it is regarded as the start of the disaster and hardship endured by Palestinians, when over 700,000 Palestinian Arabs – representing almost half of prewar Palestine's Arab population – were forced to flee or evicted from their homes and became refugees living in refugee camps across countries including Lebanon, Jordan, Syria, besides the West Bank and Gaza Strip (Manna', 2013). For the past 74 years, Palestinians have gone through, and are still going through, multiple traumatic events caused by multiple wars, including invasions, detentions, land seizures, evictions and demolitions (Veronese et al., 2014, 2017).

Palestinians were exposed to different traumatic events whether directly or indirectly, such as witnessing mutilated bodies on TV, being exposed to heavy artillery shelling, seeing evidence of shelling, hearing sonic sounds from jet fighters, witnessing killing of beloved ones and being used as a human shield by Israeli soldiers (Thabet et al., 2014, 2018). Exposure to such traumas made Palestinians susceptible to developing trauma-related symptoms or post-traumatic stress disorder (PTSD), as it has been recognized that exposure to war-related traumas will increase chances of developing PTSD, especially when someone experiences the loss of a family member, which is a common occurrence in Palestine (El-Khodary et al., 2020).

PTSD can be defined as a psychiatric disorder that develops after a critical event, a fact that has endangered the health and physical or mental integrity of the individual; it is characterized by particularly disabling symptoms such as intrusion, avoidance and changes in the mood and thoughts (Perrotta, 2019).

From a behavioral standpoint, PTSD can be explained by proposing that when someone is subjected to a life-threatening episode, they may become conditioned to different stimuli that were present during the event (sounds, time of day and smells), through the procedure of classical conditioning; thus, these formerly neutral stimuli become capable of arousing intensive anxiety.

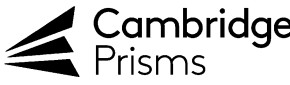



Anxiety is evoked not just by stimuli that were present during the trauma, rather via the processes of higher-order conditioning and stimuli generalization (Friedman et al., 2007). From a cognitive perspective, when someone experiences a traumatic event that disrupts their core beliefs, this can lead to negative beliefs that maintain and contribute to the development and exacerbation of PTSD symptoms (Park et al., 2012).

In a systematic review conducted by Marie et al. (2020) and targeting Palestinians living in the West Bank and Gaza Strip, the results showed that PTSD was the most common mental disorders among Palestinian children, adolescents and women. Similarly, a meta-analysis and systematic review of 28 articles in 2021 by Agbaria et al. has revealed that the prevalence of PTSD in Palestinian children and adolescents was 36%, ranging from 6% to 70%. A subgroup analysis suggested that the prevalence of PTSD did not vary based on region (West Bank, Gaza Strip).

Thabet and Vostanis (2017) have investigated the consequences of war-induced trauma on Palestinian adults in Gaza, and the results showed that 42% of them met the full criteria of PTSD. Another study has explored the prevalence and nature of war traumatic events and PTSD among Palestinian adolescents in the Gaza Strip. The results showed that most of the adolescents experienced personal trauma (N = 909, 88.4%), witnessed trauma to others (N = 861, 83.7%) and observed demolition of property (N = 908, 88.3%) during the war on Gaza, and 53.5% of the sample met DSM-V diagnostic criteria of PTSD (El-Khodary et al., 2020). Furthermore, Mahamid et al. (2023) have studied the association between political conflict and posttraumatic stress symptoms among Palestinian adults and found that there was a positive correlation between political violence and signs of trauma.

Several scales have been created in order to evaluate PTSD and trauma. For example, Hansen et al. (2010) have developed a short self-report instrument known as PTSD-8, whose validity and reliability were tested on participants from the Danish culture. In addition, Blevins et al. (2015) have revised the Posttraumatic Stress Disorder Checklist (PCL) to reflect DSM-5 changes in the PTSD criteria; the scores proved PCL-5 to be a reliable and psychometrically sound measure of PTSD symptoms. Ibrahim et al. (2018) have tested the factorial structure of PCL-5 as a screening instrument for military-related trauma among displaced Kurds. The findings proved the immense worth of PCL-5 as a screening tool to assess traumatic symptoms in Kurdish population.

There has been a shortage of instruments specific to the Palestinian region for determining traumatic symptoms. For example, in a study that explored the longitudinal association of PTSD with health-related quality of life in cardiac patients particularly in conflict-affected settings, the researchers used the PTSD Checklist Specific (PCL-S) to assess PTSD (Allabadi et al., 2021). To examine the relationship between siege stressors, war trauma and PTSD among Palestinian women in the Gaza Strip, PCL (DSM-IV) (Arabic version) was used (Aqel and Thabet, 2018).

## Current study

According to studies showing that Palestinians might meet the full criteria of PTSD because of exposure to political traumatic events (Thabet and Vostanis, 2017; El-Khodary et al., 2020; Marie et al., 2020; Agbaria et al., 2021), and because of a lack of context-specific scales to assess PTSD among Palestinian adults, this study aimed to develop a new scale and test its psychometric properties that can fully represent PTSD symptomatic expression in the context of ongoing and unresolvable conflict.

## Methodology

### *Participants and procedures*

A survey was carried out in January 2023 focusing on Palestinian adults who were living in the West Bank. Participants were recruited through online advertisements, e-mail campaigns and social media. The aims of this study, along with the procedures, were presented online. Participants responded with an e-mail expressing willingness to take part. Afterwards, every participant received a letter that briefly explained the study's subject and purpose, along with details on ethical concerns regarding confidentiality, informed consent and voluntary participation. Participants provided informed consent after carefully reading and accepting the conditions stated in the e-mail. In all 490 Palestinian adults took part – 230 males and 260 females. Participants' age ranged from 22 to 57 years (M = 34.6, SD = 13.21). Most participants (70.7%) were living in the urban regions of West Bank, and the remaining 29.3% from rural regions. Many of them (68.2%) had an academic degree, while 31.8% were non-degree holders. In order to be included in the study, participants were required to be: (1) Palestinians, (2) not to have any previous diagnosis of mental health disorders and (3) native Arabic speakers. The study was approved by An-Najah IRB before data collection began.

### *Measures*

#### *The Impact of the Event Scale (IES-R)*

IES-R is a self-report scale designed to test reactions in response to several traumatic events. The instrument consists of 22 items under three subscales: avoidance, intrusions and hyperarousal. The "intrusions" subscale represents items related to nightmares, intrusive thoughts and memories, intrusive imagery and feelings related to traumatic events. The "avoidance" subscale represents items related to avoiding places, people and things related to traumatic experiences. The "hyperarousal" subscale represents items related to anger and irritability, difficult concentration and psychophysiological arousal during exposure to reminders of traumatic experiences (Weiss, 2007).

#### *The WHOQOL-BREF*

WHOQOL-BREF is a self-report scale consisting of 26 items designed to test individual's perspectives of their life satisfaction and health over a two-week period. Individuals respond to items using a five-point Likert scale ranging from (1) not at all to (5) an extreme amount. WHOQOL-BREF is a shorter version of WHOQOL-100. Both scales were developed by the World Health Organization to explore quality of life and health among individuals with and without diseases. Individuals with high scores on WHOQOL-BREF indicated a better quality of life and higher health.

#### *Trauma scale specific to Palestinian context*

The Palestinian context-specific trauma scale (PCSTS) is a self-report measure comprising 32 items designed to test traumatic symptoms among Palestinian adults. The scale was developed based on literature related to psychometric instruments designed to test traumatic experiences (Eskin et al., 2020; Bdier et al., 2023; Mahamid, 2020). The scale consists of 32 items under five subscales:

(1) The "re-experiencing trauma" subscale represents five items related to intrusive images, thoughts, dreams and flashbacks related to traumatic experiences.

(2) The "avoidance and numbing" subscale represents nine items related to avoidance of feelings, thoughts, places, individuals and conversations related to traumatic experiences.

(3) The "hyperarousal" subscale represents six items related to insomnia, general irritability and difficulties in concentration due to traumatic experiences.

(4) The "somatic symptoms" subscale represents six items including hyperactivity, lack of energy and other somatic symptoms due to traumatic experiences.

(5) The "psychological symptoms" subscale represents six items related to psychological reactions due to traumatic experiences, such as lack of trust, low self-esteem, hopelessness, anxiety and depressive symptoms.

PCSTS is a five-point Likert scale (always = 5, mostly = 4, sometimes = 3, rarely = 2 and never = 1). A score of 1–1.8 suggests no trauma symptoms, while a score of 1.81–2.6 indicates mild trauma symptoms, 2.61–3.40 indicates moderate trauma symptoms and 3.41–4.2 indicates high trauma symptoms. Finally, a score of 4.21–5 suggests severe trauma symptoms.

Ten experts reviewed the scale for content validity and comprehensiveness; a minimum of 80% agreement between experts was fixed for each item. Certain items in the instrument were modified based on the advice from committee members. Exploratory factor analysis (EFA) and confirmatory factor analysis (CFA) were conducted to explore the stability of the structure of PCSTS.

### Data analysis

A five-factor model of PCSTS was tested through EFA and CFA using AMOS 25 software. The model showed indications of goodness of fit (GFI = .98, CFI = .96, NFI = 97, RFI = .95, RMSEA = .04, IFI = .96). Descriptive statistics were used to assess the characteristics of the scale. Independent sample *t*-test was performed to evaluate differences in traumatic symptoms among participants through demographic variables such as gender, academic status and residence, all of which apply to traumatic symptoms.

**Table 1.** Covariance of the five-factor construct (*n* = 490)

| Estimate | Standardized | SE | CR | *p* |
|---|---|---|---|---|
| Re-experiencing trauma | .428 | .112 | 3.840 | *** |
| Avoidance and numbing | .536 | .130 | 4.111 | *** |
| Hyperarousal | .527 | .097 | 5.427 | *** |
| Somatic symptoms | .544 | .102 | 5.340 | *** |
| Psychological symptoms | .597 | .131 | 4.567 | *** |

***p* significance ≤ .001.

Concurrent validity was explored by testing the correlation between IES-R, QOL and the present instrument. Finally, Guttmann split-half and Cronbach's alpha were calculated to assess the internal consistency of the scale along with test–retest reliability.

### Findings

### Exploratory factor analysis

Results of EFA showed a five-factor solution for the scale among Palestinians (see Table 1 and Figure 1); these five factors explained 69.68% of the cumulative variance. The eigenvalues of the five factors were 44.42, 8.92, 7.07, 5.55 and 3.71.

### Confirmatory factor analysis

Before proceeding to CFA, an item–total correlation was calculated using all sample data (N = 490) to ensure that all items had a strong correlation to the trauma total score (0.42–0.75). The five trauma factors were *re-experiencing trauma* (RT), *avoidance and numbing* (AN), *hyperarousal* (H), *somatic symptoms* (SS) and *psychological symptoms* (PS). The initial model assumed the trauma scale to be multidimensional and consisting of a five-factor structure. Results of CFA (see Figure 2) showed a good fit of our model in assessing

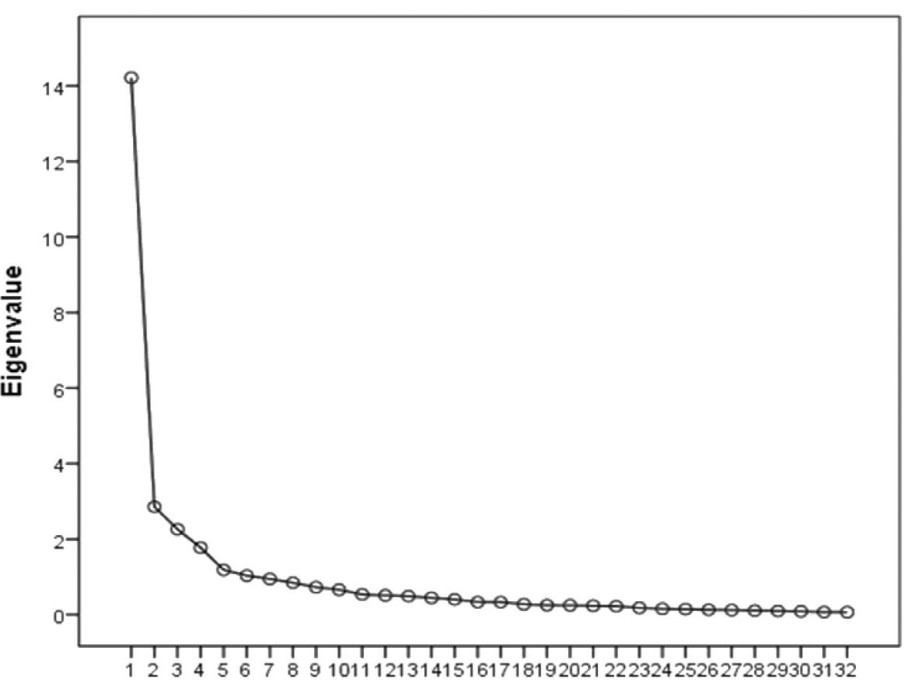

**Figure 1.** Number of factors and their eigenvalues.

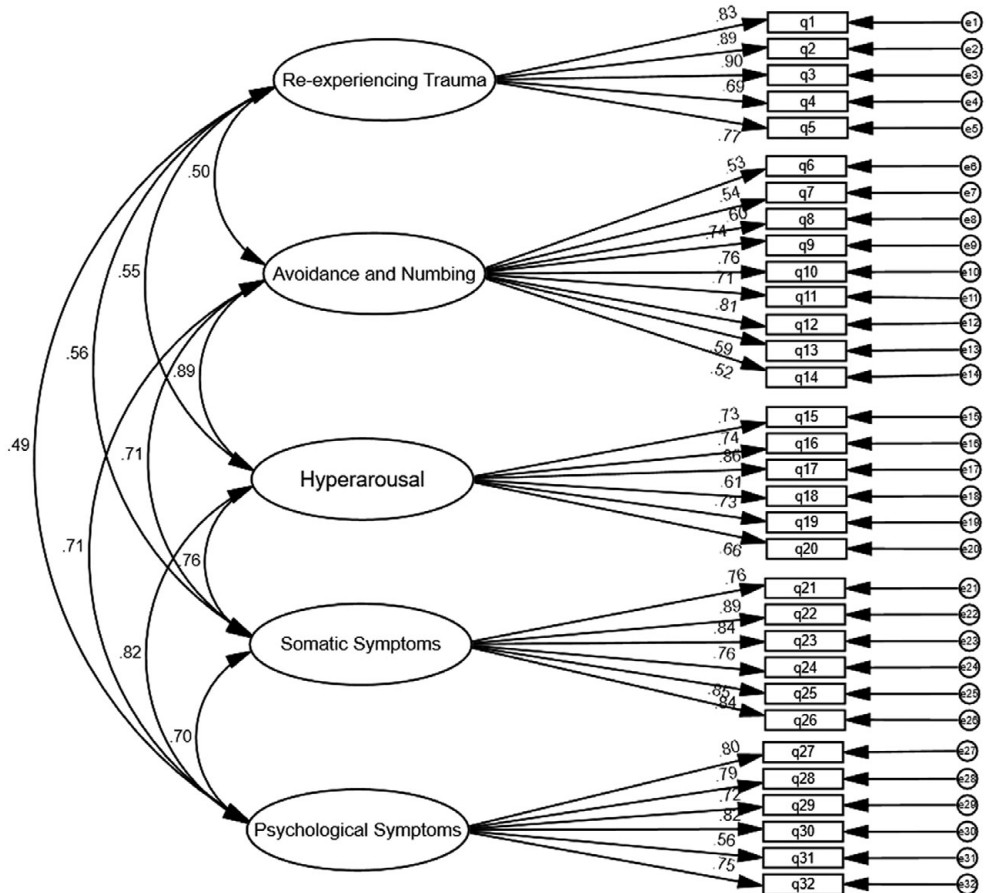

**Figure 2.** CFA of a trauma scale within the Palestinian context.

**Table 2.** Pearson correlation between trauma, IES-R and QOL scales (*n* = 490)

| Scale | 1 | 2 | 3 | 4 | 5 | 6 | 7 | 8 |
|---|---|---|---|---|---|---|---|---|
| 1. Re-experiencing trauma | — | .45** | .55** | .56** | .49** | .71** | .57** | −.32** |
| 2. Avoidance and numbing | | — | .77** | .64** | .58** | .85** | .64** | −.19** |
| 3. Hyperarousal | | | — | .71** | .72** | .90** | .69** | −.31** |
| 4. Somatic symptoms | | | | — | .63** | .85** | .72** | −.22** |
| 5. Psychological symptoms | | | | | — | .82** | .73** | −.39** |
| 6. Trauma total | | | | | | — | .81** | −.34** |
| 7. IES-R | | | | | | | — | −.36** |
| 8. QOL | | | | | | | | — |

**Correlation is significant at 0.01 level (two-tailed).

traumatic symptoms in the Palestinian context (GFI = .98, CFI = .96, NFI = 97, RFI = .95, RMSEA = .04, IFI = .96).

### *Concurrent validity*

To evaluate the concurrent validity of the trauma scale, a Pearson correlation coefficient was calculated between the trauma context, IES-R and QOL scales. Results of concurrent validity are presented in Table 2.

The findings of correlational analysis showed that RT positively correlated with AN (r = .45, *p* < .01), H (r = .55, *p* < .01), SS (r = .56,

*p* < .01), PS (r = 49, *p* < .01), trauma total (r = 71, *p* < .01), IES-R (r = 57, *p* < .01), and negatively correlated with QOL (r = −.32, *p* < .01). AN positively correlated with H (r = .77, *p* < .01), SS (r = .64, *p* < .01), PS (r = .58, *p* < .01), trauma total (r = .85, *p* < .01), IES-R (r = .64, *p* < .01), and negatively correlated with QOL (r = −.19, *p* < .01). H positively correlated with SS (r = .71, *p* < .01), PS (r = 49, *p* < .01), trauma total (r = .90, *p* < .01), IES-R (r = .69, *p* < .01), and negatively correlated with QOL (r = −.31, *p* < .01). SS positively correlated with PS (r = .63, *p* < .01), trauma total (r = .85, *p* < .01), IES-R (r = .72, *p* < .01), and negatively correlated with QOL (r = −.22, *p* < .01). PS positively correlated

**Table 3.** Reliability analysis of the trauma scale (*n* = 490)

| Item No. | Items | Cronbach's alpha if item deleted | Total correlation | Test–retest | Guttmann split-half | α |
|---|---|---|---|---|---|---|
| 1 | Images of arrests dominate my thoughts | .91 | .61 | .84 | | |
| 2 | I recall thoughts and painful memories about those wounded and/or killed | .90 | .56 | .83 | | |
| 3 | Images of soldiers' incursions penetrate my mind | .84 | .58 | .83 | | |
| 4 | I suffer from nightmares after witnessing arrests by soldiers | .90 | .61 | .82 | | |
| 5 | I cannot prevent myself from thinking of the armed clashes and/or shootings | .83 | .59 | .80 | | |
| 6 | I avoid things that remind me of night raids and/or shootings | .82 | .50 | .81 | | |
| 7 | I turn off the TV, mobile and/or radio when they broadcast news related to armed attacks | .86 | .39 | .81 | | |
| 8 | I stop myself from thinking of the storming of homes by soldiers | .91 | .53 | .82 | | |
| 9 | I feel isolated after watching the demolitions of homes by soldiers | .88 | .76 | .85 | | |
| 10 | I become lesser communicative with others after witnessing arrests made by soldiers | .90 | .66 | .83 | | |
| 11 | I become less interested in activities, which I was interested in before, after night incursions and the spread of military checkpoints | .87 | .71 | .85 | | |
| 12 | I avoid places that remind me of incidents of shootings and/or military incursions | .89 | .66 | .81 | | |
| 13 | I avoid people who remind me of painful events related to political conflict | .91 | .48 | .82 | | |
| 14 | I feel emotionless after watching assassinations and/or killings perpetrated by soldiers | .93 | .44 | .81 | | |
| 15 | I become easily distressed after experiencing daily military events | .92 | .64 | .84 | | |
| 16 | I become easily frustrated after witnessing shooting incidents and/or military incursions | .91 | .64 | .83 | | |
| 17 | I face difficulty with concentration after witnessing the storming of schools and shooting by soldiers | .90 | .76 | .84 | | |
| 18 | I have difficulty recalling painful images that I have seen | .90 | .56 | .82 | | |
| 19 | I have difficulty sleeping after the storming of our area by soldiers | .91 | .73 | .81 | | |
| 20 | I have difficulty accomplishing tasks I perform after closures and/or intrusions that our area has been subjected to | .81 | .68 | .80 | | |
| 21 | My heart beats faster when I recall images of destruction and/or killing | .87 | .75 | .83 | | |
| 22 | I feel trembling in my body when I recall images of military vehicles entering my area | .86 | .72 | .84 | | |
| 23 | I suffer from a lack of appetite when I remember painful images | .86 | .71 | .82 | | |
| 24 | I suffer from headaches when I remember painful images | .87 | .70 | .86 | | |
| 25 | I have pains in my abdomen after watching painful events | .89 | .68 | .82 | | |
| 26 | I have difficulty breathing when I recall painful images | .79 | .69 | .84 | | |
| 27 | I feel hopeless after witnessing military incursions | . 81 | .59 | .81 | | |
| 28 | I feel helpless after experiencing painful events | .80 | .60 | .82 | | |
| 29 | I lose trust in others after witnessing painful events | .82 | .69 | .83 | | |
| 30 | I feel terrified after experiencing painful events | .83 | .74 | .81 | | |
| 31 | I feel sad after experiencing military invasions and/or incursions | .84 | .46 | .82 | | |
| 32 | I feel worried about the future after the current events in this political climate | .83 | .66 | .84 | | |
| | *Re-experiencing trauma* | | | .87 | .87 | .91 |
| | *Avoidance and numbing* | | | .85 | .86 | .92 |
| | *Hyperarousal* | | | .86 | .88 | .93 |
| | *Somatic symptoms* | | | .84 | .85 | .91 |
| | *Psychological symptoms* | | | .83 | .84 | .94 |
| | *Trauma total* | | | .86 | .90 | .95 |

with trauma total (r = .82, *p* < .01), IES-R (r = .73, *p* < .01), and negatively correlated with QOL (r = −.39, *p* < .01). In addition, trauma total positively correlated with IES-R (r = .81, *p* < .01) and negatively correlated with QOL (r = −.34, *p* < .01). Finally, IES-R negatively correlated with QOL (r = −.34, *p* < .01).

### Reliability of the scale

To test the reliability of trauma context scale, test–retest, Cronbach's alpha and Guttmann split-half were calculated as shown in Table 3.

The results of Cronbach's alpha showed a high level of reliability (α = .95). In addition, split-half yielded a high level of internal reliability (.90). Test–retest was calculated by re-administering the scale on 100 participants 3 weeks after the first administration. The correlation between the first and the second was 0.86, which shows that our trauma context scale is reliable in assessing traumatic symptoms in the Palestinian context.

The results of independent samples t-test showed significant differences in traumatic symptoms between males and females, and in favor of females (t = 9.15, *p* < .01). Significant differences were found in trauma symptoms between academic degree holders and non-degree holders, in favor of non-degree holders (t = 7.64, *p* < .01). Finally, significant differences were noted between urban and rural residents (t = 8.43, *p* < .01), in favor of urban residents.

### Discussion

The current study aimed to develop an instrument to evaluate traumatic symptoms in the Palestinian context. Our findings showed that PCSTS is a valid and reliable scale for assessing traumatic symptoms. They also revealed a positive association between PCSTS and the IES-R scales. In addition, a negative association was found between QOL and PCSTS, which indicates the soundness and trustworthiness of our scale. CFA confirmed a stable construct of the five-factor structure of PCSTS: (1) re-experiencing trauma, (2) avoidance and numbing, (3) hyperarousal, (4) somatic symptoms and (5) psychological symptoms. Research has indicated the importance of these domains in assessing traumatic symptoms among different populations. LeBeau et al. (2014) have evaluated the factorial structure of the NSESSS-PTSD scale among clinical and non-clinical groups, which revealed multiple dominant constructs of the scale: avoidance, re-experiencing, negative alterations in cognitions or mood and hyperarousal. Gootzeit et al. (2015) have developed the Iowa Traumatic Response Inventory to evaluate symptoms related to PTSD, whose findings indicated that intrusions and avoidance are positively correlated and are specific to PTSD. Wigham et al. (2011) have developed the Lancaster and Northgate Trauma Scales (LANTS) to test traumatic life events, whose results indicated that LANTS is a valid instrument for assessing PTSD symptoms (Table 4).

The long-standing Israeli–Palestinian conflict continues to manifest negative mental health outcomes among Palestinians.

Palestinians living in East Jerusalem, West Bank and Gaza Strip are at risk of developing mental health problems due to the prolonged traumatic and stressful experiences. The Israeli military occupation of West Bank and Gaza Strip has resulted in continued threats of beating, arrests and possible home demolitions. In addition, the spread of checkpoints and road closures in the West Bank limited the mobility of Palestinians (Mahamid, 2020). Therefore, there is pressing need for assessing trauma and related disorders among Palestinians.

Developing new tools to diagnose traumatic symptoms in the Palestinian society will help improve mental health care for people who are experiencing the most tragic acts including demolitions of houses, confiscation of land, arrests, pursuits and other forms of violence (Banat et al., 2018).

Our findings showed significant differences in trauma symptoms between males and females, in favor of male participants. As a characteristic of Palestinian society, women engage too little with activities outside of home, while cases of torture and murder perpetrated by assailants were rampant and most of the victims comprised men. It is therefore possible that males may have experienced more political trauma than females. These findings are similar to Yasan et al. (2009), who found that men experiencing military conflicts are more at risk of developing trauma than women.

Regarding academic status, our study revealed that non-degree holders reported more traumatic symptoms than degree holders. This supports the research conducted in northeast Iran by Modaghegh et al. (2013), who found that the level of trauma was greater among uneducated general population and primary-educated population compared to educated people. One possible explanation for this result is that education could improve people's abilities to deal with traumatic experiences. Education can stimulate resilience, nurture learners' social and emotional development, give individuals and communities hope for the future, and in the long term encourage social cohesion, reconciliation and peacebuilding (Morton, 2018).

Our study showed that urban residents reported greater traumatic behaviors than rural residents. In the last few months, most urban Palestinian regions, including displacement camps, witnessed several political traumatic events, represented by night raids, house demolitions, killings and arrests by the Israeli security forces. Our results are in line with other studies showing a higher prevalence of traumatic symptoms among adolescents living in Palestinian cities compared to those living in towns and villages (Giacaman et al., 2007).

### Limitations of the study

The construction of psychological tools is a continuing procedure. First, our study used convenience and snowballing sampling to recruit Palestinian adults via online self-reports. It is paramount to evaluate the trauma scale with various cohorts, particularly those exposed to critical traumatic occurrences. Second, we developed our scale during a difficult period characterized by high levels of

**Table 4.** Differences in trauma symptoms by academic status, gender and residence (*n* = 490)

| Dependent variable | Variable | Gender | | Academic status | | Residence | |
|---|---|---|---|---|---|---|---|
| | | Female | Male | Degree holders | Non-degree holders | Urban | Rural |
| Trauma total | Mean (SD) | 2.59 (.72) | 3.21 (.74) | 2.70 (.69) | >3.12 (.79) | 3.33 (.64) | 2.64 (.82) |

political violence, arrests and nighttime incursions into various areas of the West Bank. The political conflict may have augmented trauma symptoms among Palestinians, which could have impacted the factor structure of the scale. Further research is necessary to assess the psychometric characteristics of the trauma scale at different points in time. Third, the sample is not totally representative of different Palestinians groups, such as battered women, lower socio-economic groups, former prisoners and victims of political conflicts. Therefore, it is recommended to test the psychometric properties of the trauma scale with specific groups, especially victims of political violence and those at risk of exposure to political traumas. Finally, the concurrent validity of the trauma scale was tested against different scales but whose validation within the present Palestinian context is yet to be determined.

## Conclusions

The present study aimed to develop and validate a context-specific trauma scale. It showed a stable five-factor construct of PCSTS in assessing traumatic symptoms among Palestinian adults: (1) re-experiencing trauma, (2) avoidance and numbing, (3) hyper-arousal, (4) somatic symptoms and (5) psychological symptoms. Moreover, a positive association was found between IES-R and PCSTS, confirming the latter's validity. Developing and validating new instruments to test traumatic symptoms in the Palestinian context will help health providers develop several interventions targeting individuals and groups who are at risk of trauma and related disorders.

**Open peer review.** To view the open peer review materials for this article, please visit http://doi.org/10.1017/gmh.2023.82.

**Data availability statement.** The datasets generated during and/or analyzed during the current study are available from the corresponding author on reasonable request and are also available in the OSF repository (https://osf.io/xw87t/files/osfsstorage).

**Author contribution.** All authors contributed equally to this article, Dr. Bdier prepared the literature review section, Dr. Mahamid prepared methodology and analysis sections. Finally, Dr. Veronese prepared the discussion section.

**Financial support.** No funding was received for this study.

**Competing interest.** The authors declare none.

**Ethics approval and consent to participate.** All procedures involving human participants were in accordance with the ethical standards of An-Najah National University IRB, the American Psychological Association (APA, 2010) and the Helsinki Declaration (2013). Informed consent was obtained from all participants. The protocol of our study received ethical approval from An-Najah National University IRB.

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
