## [Reviewer Report]

Dear Editors, 

We are glad to submit our paper entitled “Development of a context-specific trauma scale among a Palestinian adult population living amidst military violence”. This is an important study testing the development of a context-specific trauma scale in a context characterized by a high level of stress and ongoing trauma. Hoping the manuscript can be interesting for the readers of Global Mental Health Journal. All authors declare no conflicts of interest and agree for submitting to the journal.

Thank you for your gentle understanding, 

Yours Sincerely, 

Fayez Mahamid (on behalf of the co-authors).

---

## [Reviewer Report]

The paper is very interesting, written in a very clear, concise and outlined manner. It deals with a crucial issue of international interest. 

I have just a few minor comments:

- Method

Authors stated “The study sample was recruited using online tools”. Can you describe it better?

Moreover, there are no information regarding the age of your participants.

Can you provide Mean, SD?

p.3, line 26: check the language. ‘the Marie et al literature review”

p.4 line 55 – check references

p.12, line 47 cut the sentence. …. . Moreover,…

p.14. Discussing the difference among educated and less educated people, maybe you can add a discussion about class and economic factors? It could be a factor that have a crucial part in diminishing feeling of insecurity or enhancing it.

Maybe you can discuss about it in the limitation section

---

## [Reviewer Report]

Thank you for a good article. Please review the references once again for APA formatting.

otherwise, a well-written and appropriate article. This scale is much needed in the context. congratulations.

---

## [Reviewer Report]

Dear Editors, 

We are glad to submit our paper entitled “Development of a context-specific trauma scale among a Palestinian adult population living amidst military violence”. This is an important study testing the development of a context-specific trauma scale in a context characterized by a high level of stress and ongoing trauma. Hoping the manuscript can be interesting for the readers of Cambridge Prisms: Global Mental Health Journal. All authors declare no conflicts of interest and agree for submitting to the journal.

Thank you for your gentle understanding, 

Yours Sincerely, 

Fayez Mahamid (on behalf of the co-authors).

---

## [Reviewer Report]

I thank the authors for listening to my suggestions and integrating them into the text. good work!